# Multi-dimensional oscillatory activity of mouse GnRH neurons in vivo

Su Young Han[1], Shel-Hwa Yeo[1], Jae-Chang Kim[2], Ziyue Zhou[1], Allan E Herbison[1]*

[1]Department of Physiology, Development and Neuroscience, Downing site, University of Cambridge, Cambridge, United Kingdom; [2]Zurich Center for Neuroeconomics, Department of Economics, University of Zurich, Zurich, Switzerland

## eLife Assessment

This **valuable** study investigates the oscillatory activity of gonadotropin-releasing hormone (GnRH) neurones in mice using GCaMP fiber photometry. It demonstrates three distinct patterns of oscillatory activity that occur in GnRH neurons comprising low-level rapid baseline activity, abrupt short-duration oscillations that drive pulsatile gonadotropin secretion, and, in females, a gradual and prolonged oscillating increase in activity responsible for the relatively short-lived preovulatory LH surge. The evidence presented in the study is **solid**, offering theoretical implications for understanding the behaviour of GnRH neurones in the context of reproductive physiology, and will be of interest to researchers in neuroendocrinology and reproductive biology.

*For correspondence:
aeh36@cam.ac.uk

Competing interest: The authors declare that no competing interests exist.

**Abstract** The gonadotropin-releasing hormone (GnRH) neurons represent the key output cells of the neural network controlling mammalian fertility. We used GCaMP fiber photometry to record the population activity of the GnRH neuron distal projections in the ventral arcuate nucleus where they merge before entering the median eminence to release GnRH into the portal vasculature. Recordings in freely behaving intact male and female mice revealed abrupt ~8 min duration increases in activity that correlated perfectly with the appearance of a subsequent pulse of luteinizing hormone (LH). The GnRH neuron dendrons also exhibited a low level of unchanging clustered, rapidly fluctuating baseline activity in males and throughout the estrous cycle in females. In female mice, a gradual increase in basal activity that exhibited ~80 min oscillations began in the afternoon of proestrus and lasted for 12 hr. This was associated with the onset of the LH surge that ended several hours before the fall in the GCaMP signal. Abrupt 8 min duration episodes of GCaMP activity continued to occur on top of the rising surge baseline before ceasing in estrus. These observations provide the first description of GnRH neuron activity in freely behaving animals. They demonstrate that three distinct patterns of oscillatory activity occur in GnRH neurons. These are comprised of low-level rapid baseline activity, abrupt 8 min duration oscillations that drive pulsatile gonadotropin secretion, and, in females, a gradual and very prolonged oscillating increase in activity responsible for the preovulatory LH surge.

## Introduction

The founding concepts of reproductive neuroendocrinology arose from the recognition by Harris and colleagues of the neurohumoral basis of the hypophyseal pituitary system (**Harris, 1955**; **Fink, 2015**) and the subsequent discovery by Schally and Guillemin and co-workers of the decapeptide GnRH (**Amoss et al., 1971**; **Schally et al., 1971**). This was soon followed by an appreciation that GnRH neurons had a very unusual topography in which cell bodies scattered throughout the basal forebrain sent projections that converged on the median eminence to release GnRH into the portal system

(*Barry, 1979*). The quest to understand the patterns of GnRH released into the portal system was largely solved by the portal bleeding approach in sheep (*Clarke and Cummins, 1982*; *Caraty and Locatelli, 1988*). Notably, this revealed that abrupt increments of GnRH secretion evoked pulsatile gonadotropin secretion whereas, in females, prolonged increases in constant or episodic GnRH secretion were responsible for the preovulatory LH surge (*Karsch et al., 1997*; *Clarke, 2018*).

Establishing the activity patterns of GnRH neurons in vivo has been hampered considerably by their widely dispersed topography and, five decades on, remains an elusive goal (*Herbison, 2015*; *Constantin, 2017*). The development of transgenic GnRH-GFP mouse lines greatly facilitated efforts to record the electrical properties of individual GnRH neurons in the acute brain slice (*Constantin et al., 2021a*). However, the patterns of firing exhibited by GnRH neurons under these conditions did not align well with predicted pulse or surge patterns of GnRH secretion (*Herbison, 2015*; *Constantin et al., 2021b*). In particular, the only report of GnRH neuron firing in vivo to date described a range of irregular firing patterns inconsistent with known hormone secretion (*Constantin et al., 2013*).

The episodic pattern of GnRH secretion driving pulsatile LH release is generated by the intermittent synchronised activity of the arcuate kisspeptin neuron population that projects to and controls the distal processes of GnRH neurons (*Herbison, 2018*; *Goodman et al., 2022*). These processes, termed dendrons in rodents, converge in the ventrolateral aspects of the arcuate nucleus (ARN) before turning into short axons that run into the median eminence (*Herbison, 2021*). The GnRH neuron dendron appears to operate as an autonomous site for GnRH pulse secretion whilst also carrying action potentials driving the GnRH surge that are initiated upstream at the level of the GnRH neuron cell bodies (*Herbison, 2020*).

We have previously documented that the expression of GCaMP in GnRH neuron cell bodies and dendrons can be used to faithfully record GnRH neuron electrical activity in vitro (*Iremonger et al., 2017*). As the ventrolateral ARN provides a unique location of concentrated GnRH neuron dendrons,

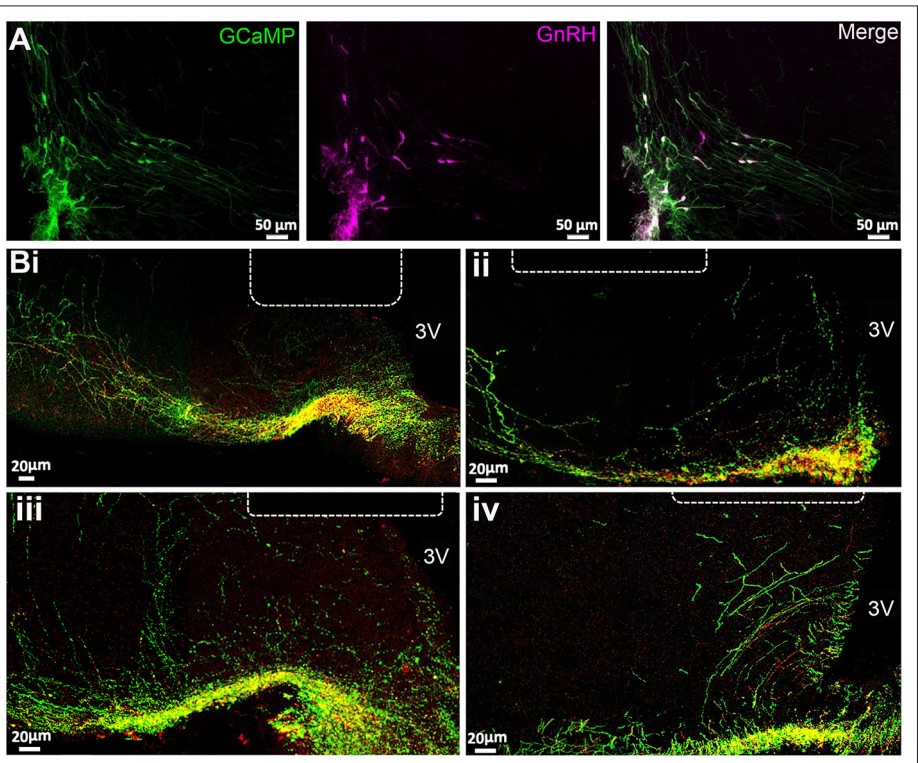

**Figure 1.** GCaMP expression in gonadotropin-releasing hormone (GnRH) neurons. (**A**) Photomicrographs at the level of organum vasculosm of the laminae terminalis (OVLT) showing GFP immunofluorescence representing GCaMP6 (green), GnRH (purple), and merged channels showing double labeled cells (white). (**B**) Photomicrographs from four mice at the level of arcuate nucleus showing the location of the end of the optic fiber (dotted line) in relation to GnRH neuron dendrons labeled with GnRH (red), GFP representing GCaMP6 (green) in merged channels where double labeled elements are yellow. 3 V, third ventricle.

we reasoned that it may be possible to record their activity in freely behaving mice at this location using GCaMP fiber photometry. We demonstrate that this is indeed possible and describe here the patterns of GnRH neuron activity underlying episodic and surge patterns of hormone secretion in male and female mice.

## Results

### Characterization of Gnrh1-GCaMP mouse

The *Gnrh1*$^{Cre/+}$ mouse line (JAX stock #021207) (*Yoon et al., 2005*) was crossed with the Ai162 (TIT2L-GC6s-ICL-tTA2)-D Cre-dependent GCaMP6s line (JAX stock #031562) (*Daigle et al., 2018*) to generate Gnrh1-GCaMP6 mice. Dual immunofluorescence in four female mice demonstrated that 90 ± 2% of GnRH neuron cell bodies expressed GCaMP within the rostral preoptic area (*Figure 1A*) as did the majority of GnRH dendrons and fibers within the ME (*Figure 1B*). In successfully recorded mice, optic fibers were found to be located immediately above or within the mid-caudal ARN (*Figure 1B*). Mice with unsuccessful recordings (n=4), that displayed no variation in GCaMP signal over 24 hr recordings, were found to have optic fibers located either mostly within the third ventricle or >200 μM lateral to the ARN.

### GnRH neuron activity in male mice

The GnRH neuron population activity at the level of distal dendron was measured for 24 hr starting between 10.00–11.00 am in freely behaving Gnrh1-GCaMP6 male mice (n=7). Mice exhibited intermittent, abrupt increases in GCaMP signal termed here 'dendron-synchronization episodes' (dSEs; *Figure 2A*). These episodes had a total duration of ~10 min (658 ± 5 s) comprised of a rapid increase (full width at half maximum (FWHM) value of 30 ± 4 s) and a slower decline (FWHM 102 ± 7 s) (*Figure 2C*). The dSEs occurred on average every 93.3 ± 7.4 min but had a large inter-interval range of 3.9–346.6 min. The inter-peak interval distribution was right-skewed (Skewness=1.39, W=0.87, p<0.0001, Shapiro-Wilk test) and showed no clear modal pattern (Kurtosis=1.81, indicating that the data is more spread out than a normal distribution which has a kurtosis of 3.0; *Figure 2D*). The 24 hr recordings also revealed a low amplitude, higher frequency baseline pattern activity (*Figure 2A*). Control recordings from mice with misplaced fibers exhibited almost no baseline activity (*Figure 2B*).

The characteristics of dSEs strongly resembled those exhibited by arcuate nucleus kisspeptin neurons that drive pulsatile LH secretion (*Han et al., 2019*). Therefore, we assessed the relationship of GnRH neuron dSEs to LH pulses by taking 5- to 10 min tail-tip blood samples for up to 240 min periods while monitoring GCaMP fluorescence. This showed a perfect association between dSE occurrence and LH pulses (*Figure 2E*). Analysis of the temporal relationship between dSEs and LH pulses revealed that the peak of dSE consistently preceded LH pulses by 5.0 ± 0.9 min (N=6 episodes in 5 mice; *Figure 2F*).

### GnRH neuron activity in female mice

To examine patterns of GnRH neuron activity across the estrous cycle, female Gnrh1-GCaMP6 mice were recorded for 6 hr periods between 10 am-4 pm on each day of the cycle determined by vaginal cytology on the morning of the recording. Similar to males, freely behaving female mice exhibited a variable baseline pattern of high-frequency activity upon which intermittent, abrupt increases in GCaMP signal occurred (*Figure 3A*). Each dSE had a total duration of ~8 min (463 ± 22 s) with a rapid increase (FWHM 31 ± 3 s) and a slower decline (FWHM 81 ± 6 s) (*Figure 3B*). The frequency of dSEs varied across the cycle in the same manner as kisspeptin neuron SEs (*McQuillan et al., 2019*) with a dramatic slowing during estrus (dSE interval of 151.0 ± 35.3 min; p=0.01 vs diestrus and 0.001 vs proestrus, Dunn's posthoc; n=5) (*Figure 3C*). The inter-dSE intervals for metestrus, diestrus, and proestrus were not significantly different with mean ± SEM of 70.2 ± 6.7 min (n=8), 46.2 ± 4.2 min (n=8), and 38.8 ± 3.4 min (n=6), respectively (*Figure 3C*). The pattern of the inter-peak interval distribution of dSEs interval was determined in each estrous cycle. In all cycles, the distribution pattern failed the normality test (metestrus [W=0.92, p=0.0007], diestrus [W=0.88, p<0.0001], proestrus [W=0.92, p=0.0007], estrus [W=0.94, p=0.03]). In metestrus, diestrus, and proestrus, the distribution was right-skewed (skewness = 0.83, 1.37, and 1.13, respectively) with diestrus showing the most pronounced skew (*Figure 3*.Aii). In metestrus, most dSEs occurred within an interval range of 46–55 min (17.5%),

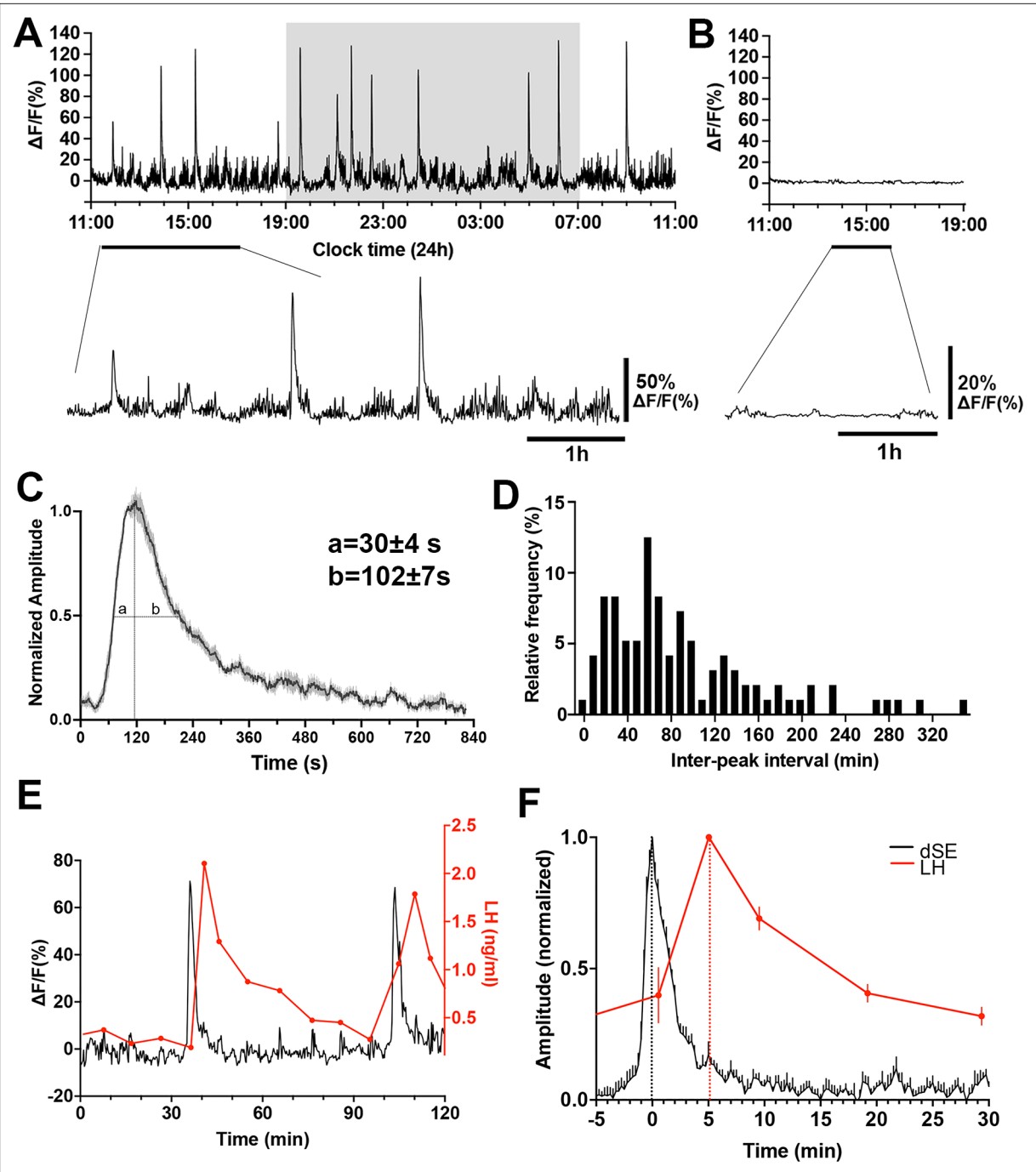

**Figure 2.** Gonadotropin-releasing hormone (GnRH) neuron dendron activity in male mice. (**A**) A representative example of 24 hr GCaMP6 photometry recording showing abrupt dendron synchronized episodes (dSEs) occurring on lower amplitude baseline activity in a male mouse. Below, an expanded view of the trace indicated by the horizontal line. (**B**) Example of 8 hr GCaMP6 photometry recording in a male mouse with a misplaced fiber optic. All recorded signals are under 3% of baseline. Below, an expanded view of the trace indicated by the horizontal line. (**C**) Average high-resolution profile of a dSE in male mice (N=7 mice) showing a rapid onset followed by a gradual decrease back to baseline. 'a' and 'b' gives values at full-width half maximum (FWHM). (**D**) Inter-peak intervals combined from all recordings (n=96 dSEs) displayed as a percentage of all intervals occurring in 10 min bins. W=0.87, p<0.0001****, Shapiro Wilk normality test. Skewness=1.39, Kurtosis=1.81. (**E**) Representative example showing the relationship of dSEs (black) to pulsatile luteinizing hormone (LH) secretion (red). (**F**) Normalized increase in LH plotted against dSEs, with the time 0 being the peak of dSE.

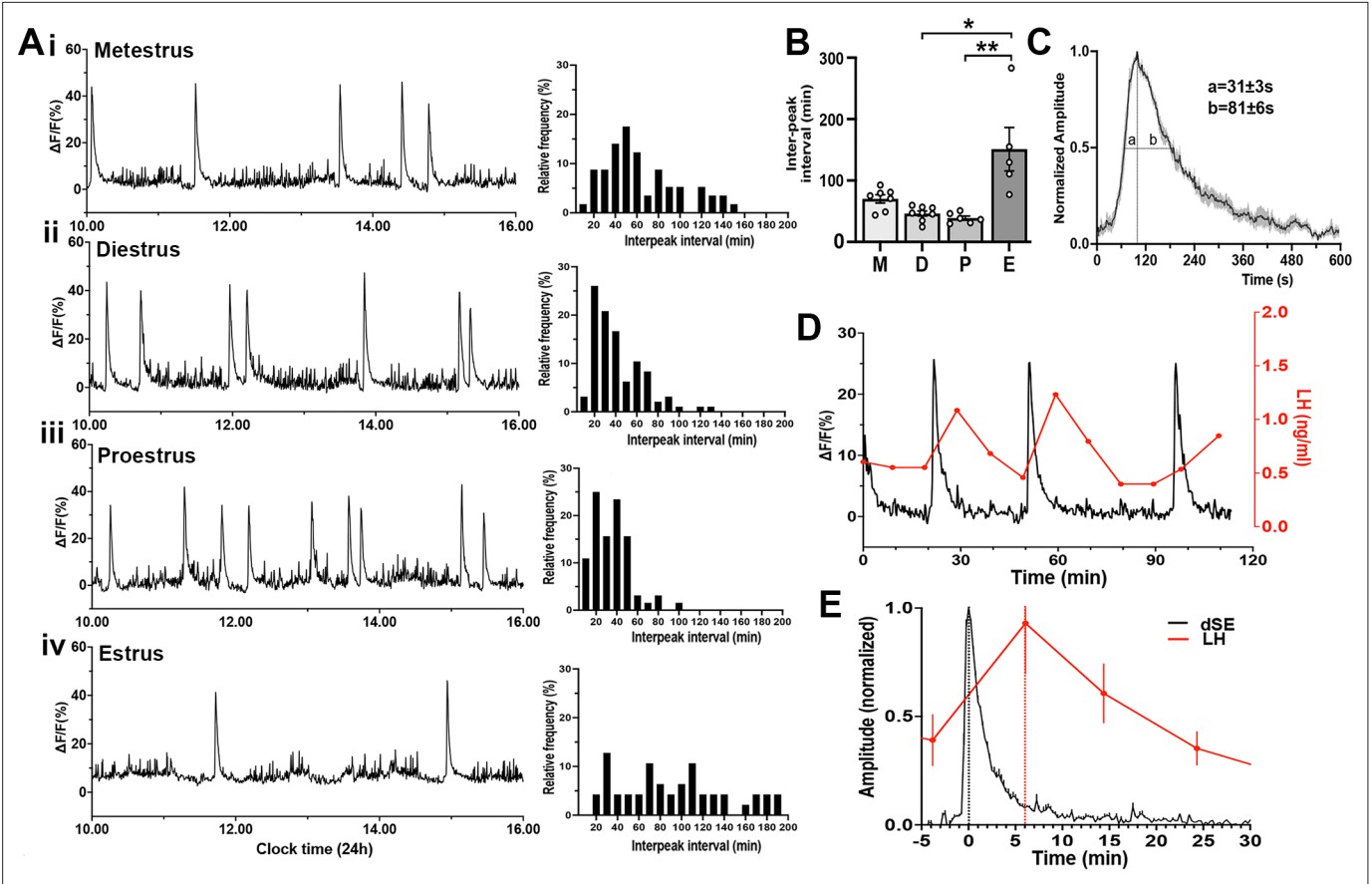

**Figure 3.** Gonadotropin-releasing hormone (GnRH) neuron dendron activity in female mice. (**A**) Representative example of 6 hr GCaMP6 photometry recordings from a female mice in (**i**) metestrus, (ii) diestrus, (iii) proestrus, and (iv) estrus. To the right of each trace are histograms showing the mean inter-peak intervals for dendron synchronized episodes (dSEs) from all recordings across metestrus (n=53 dSEs), diestrus (n=88 dSEs), proestrus (n=62 dSEs), and estrus (n=42 dSEs) displayed as a percentage of all intervals occurring in 10 min bins. (**B**) Histograms showing inter-peak intervals of dSEs across the estrous cycle. M=metestrus (N=8), D=diestrus (N=8), P=proestrus (N=6), E=estrus (N=5). *p<0.05, **p<0.01, Kruskal Wallis followed by Dunn's post-hoc test. (**C**) Average high-resolution profile of dSEs in female mice showing a rapid onset followed by a gradual decrease in the signal. 'a' and 'b' show the values at full-width half maximum (FWHM). (**D**) A representative example showing the relationship of dSEs (black) to pulsatile LH secretion (red). (**E**) Normalized change in LH (red) plotted against normalised dSEs, with the time 0 being the peak of the dSE.

while in diestrus and proestrus, the most frequent range was 16–25 min (26.0% and 25.0%, respectively)(*Figure 3A*). In contrast, during estrus, the inter-peak distribution pattern was nearly symmetric (skewness=0.59) with a flatter and broad shape, showing no clear modal distribution (kurtosis=−0.41) (*Figure 3D*). The kurtosis values were –0.22, 2.01, 1.84, and –0.41 in metestrus, diestrus, proestrus and estrus, respectively. A kurtosis of 3.0 indicates normally distributed data, suggesting that the inter-peak intervals in diestrus and proestrus have a moderate peak, while the negative kurtosis values of estrus in particular indicate a flatter and broader distribution.

To examine the relationship of dSEs to LH pulses, tail-tip blood sampling was performed and dSEs were found to have a perfect correlation with LH pulses (*Figure 3D*) with the peak of dSEs occurring 6.1 ± 0.2 min (N=6 episodes in 4 mice) before the peak of the following LH pulse (*Figure 3E*).

## Low amplitude, clustered baseline activity in GnRH neurons in both male and female mice

A low-amplitude, high-frequency GCaMP signal was observed in both male and female mice (*Figure 2A*, *Figure 3A*, *Figure 4A and B*). This activity occurred in a cluster-like manner with mean cluster durations of 22.5 ± 2.8 min (males), 30.0 ± 3.9 min (metestrus), 35.7 ± 6.0 min (diestrus) 24.8 ± 2.2 min (proestrus) and 35.3 ± 6.8 min (estrus) in female mice (*Figure 4C*). The intra-cluster frequency (range of 0.006–0.018 Hz) was not different between males and females (*Figure 4D*). Neither the

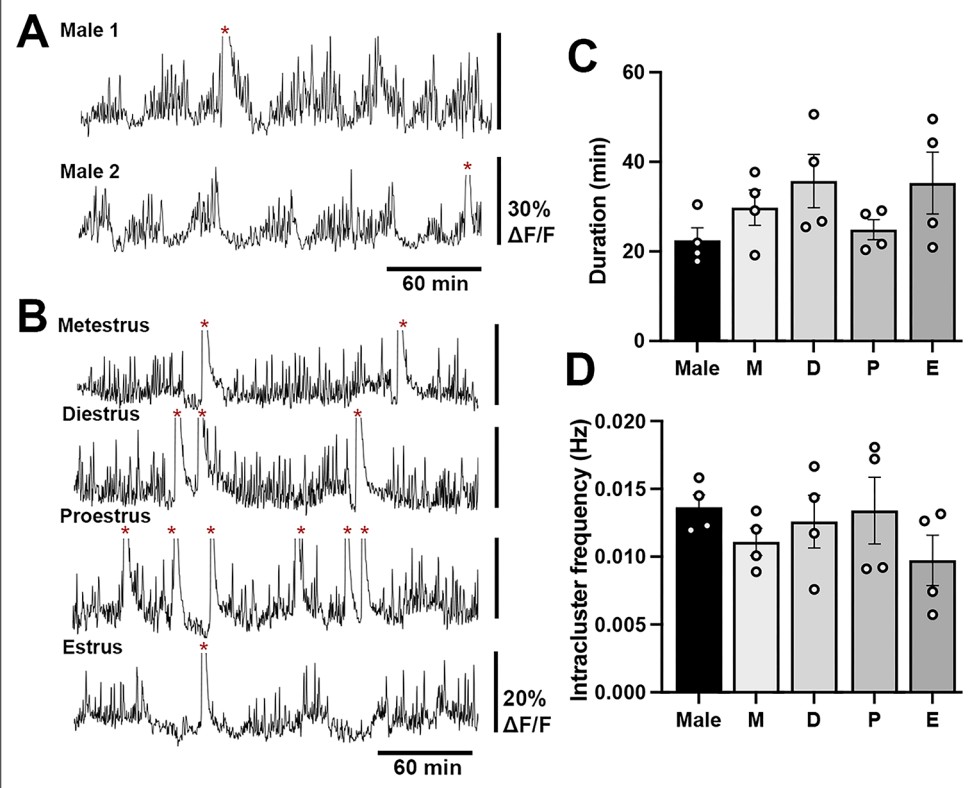

**Figure 4.** Baseline high-frequency cluster activity of gonadotropin-releasing hormone (GnRH) neuron dendrons in male and female mice. Representative 4 hr GCaMP recordings in (**A**) two male mice and (**B**) a female mouse across the different stages of the estrous cycle showing high-frequency baseline activity. The large increases in calcium activity representing dendron synchronization episodes (dSEs) are indicated by red asterisks and cut off to optimize the display of the low amplitude activities. Histograms showing the duration (**C**) and the intra-cluster frequency (**D**) of the baseline activity in male and female mice. No significance was found across the groups. M=metestrus, D=diestrus, P=proestrus, E=estrus.

mean duration nor intra-cluster frequency of the baseline activity were different between males and females or across the estrous cycle (*Figure 4C and D*). The amplitude of the baseline signal was 10–12% (males) and 5–12% (females) of the mean amplitude of dSEs.

## GnRH neuron activity in proestrous female mice

To examine patterns of GnRH neuron activity at the time of the preovulatory surge, female Gnrh1-GCaMP6 mice (n=7) were recorded for a 24 hr period beginning on the morning of proestrus. Initially, the signals were identical to those observed on metestrus and diestrus with a low level of high-frequency baseline activity upon which abrupt, short increases in activity occurred (*Figures 3A and 5A*). A gradual increase in baseline activity was observed to begin 4.0 ± 0.5 hr (range 2–6.5 hr) before 'lights-off' (19:00 hr, 12:12 lighting) that peaked 5.9 ± 0.5 hr later and declined to baseline by 12.6 ± 0.8 hr (range: 10.3–15.0 hr)(*Figure 5A*). The FWHM period of incline and decline of this slow activity was 3.1 ± 0.4 and 4.7 ± 0.5 hr, respectively. The slow increment in calcium signal was not smooth but consisted of multiple slow oscillations (*Figure 5A*). The mean duration of these slow oscillations was 78 ± 4 min (range 32–150 min). In addition, dSEs were observed to occur superimposed upon the slow oscillating signal until around the time of the peak at which time they stopped for several hours (*Figure 5A*). Similar duration 24 hr recordings from metestrus-diestrous female mice recorded the presence of dSEs without any baseline shifts.

To assess the relationship of the slow oscillating signal to the LH surge, tail-tip bleeding with a 3 hr interval was undertaken (*Figure 5B*). This showed that the LH surge began at the time of baseline GCaMP change and peaked at 9.3 ± 2.8 ng/ml close to when baseline GCaMP level was at its

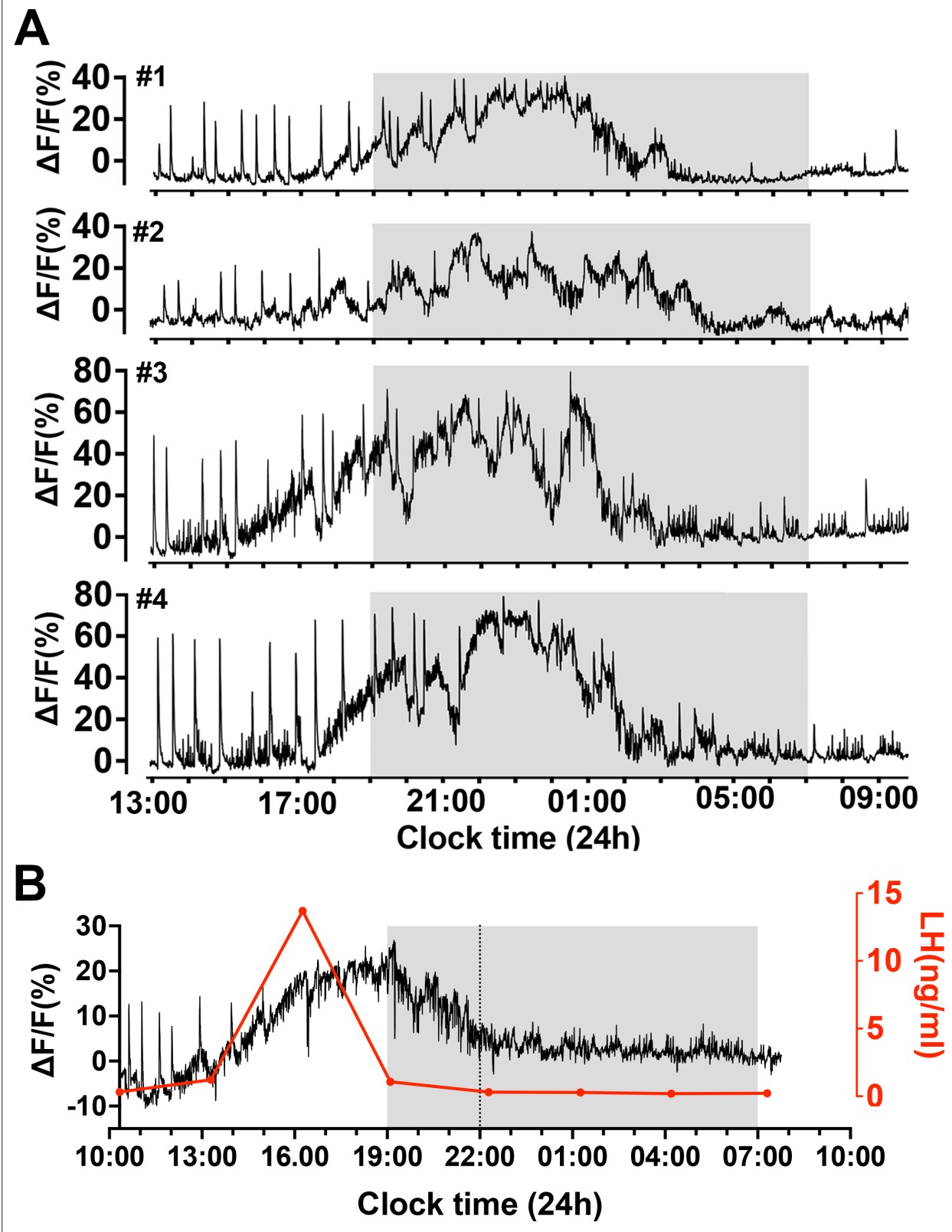

**Figure 5.** Slow oscillating increases in calcium activity on the afternoon of proestrus in female mice. (**A**) Examples of 21 h GCaMP recordings from four proestrous female mice. Note the prolonged (>10 hr) gradual oscillating increase in calcium signal beginning on the afternoon of proestrus. The sharp dendron synchronized episodes (dSEs) continue until approximately the plateau phase of the increased baseline at which time they stop or slow. (**B**) A representative example showing the relationship between the slow increase in baseline calcium activity (black) with luteinizing hormone (LH) surge (red). Note that the rise in LH occurs alongside the initial rise in calcium activity but returns to baseline several hours before the calcium signal.

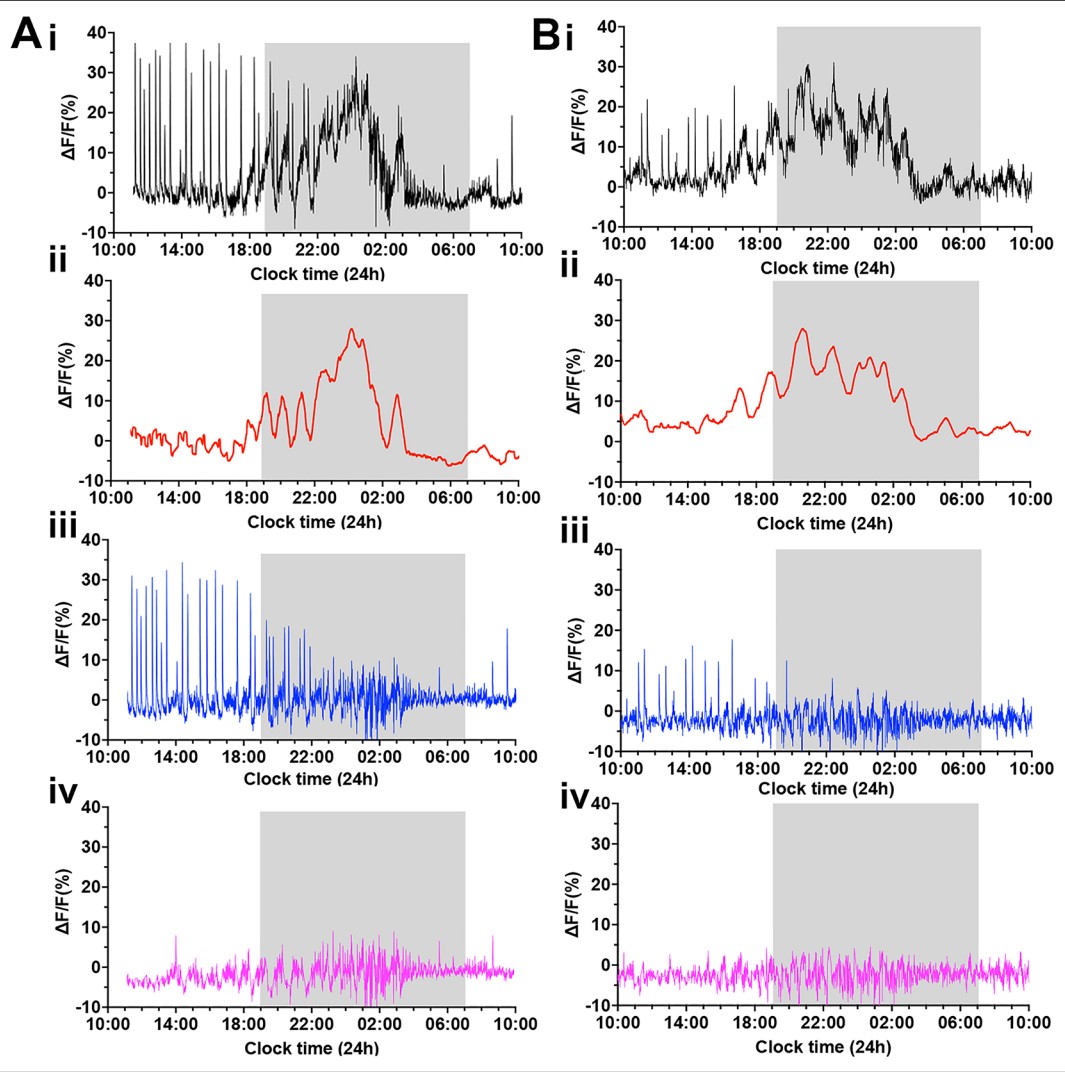

**Figure 6.** Deconvolution of the GCaMP signal across the proestrous surge reveals multimodal activity patterns of gonadotropin-releasing hormone (GnRH) neuron dendron. (**A–B**) Representative examples of 24 hr photometry recordings from two female mice starting at proestrus showing (**i**) the original recording, (ii) a 30 min rolling average highlighting the luteinizing hormone (LH) surge signal (red), (iii) the LH surge signal subtracted from the original recording, displaying the dendron synchronized episodes (dSEs) (blue), and (iv) the residual baseline signal (pink) after subtracting both the surge and pulse profiles.

highest (ΔF/F of 20–40%) (n=4). Notably, GCaMP activity greatly outlasted the duration of the LH surge (*Figure 5B*).

To clarify the different components of GCaMP activity occurring on proestrus, we used a customized Matlab code to separate out the three signals occurring at this time (*Figure 6*). First, a 30 min rolling average was used to extract the slow oscillatory rise and decline in GCaMP activity (*Figure 6ii*). The subtraction of this 'surge signal' from the original signal (*Figure 6*) shows the stereotypical dSEs and baseline fluctuations more clearly (*Figure 6iii*). Next, the peaks of individual dSEs were selected, and the signals from 60 s before and 360 s after each peak of dSE were extracted. The remaining signals are considered to be 'the residual signal' comprising the high-frequency basal activity described above (*Figure 6iv*).

## Discussion

We reveal here three distinct oscillatory patterns of GnRH neuron activity in freely behaving male and female mice. This includes a baseline of high-frequency cluster-like activity upon which abrupt ~8 min episodes occur in both males and females, and a prolonged, slowly oscillating rise in activity on the evening of proestrus in females. These patterns of GnRH neuron activity are sufficient to explain the pulse and surge patterns of LH secretion in females and pulses in males.

We demonstrate that the GCaMP fiber photometry approach used previously for assessing cell body population activity (*Han et al., 2019*; *McQuillan et al., 2019*) can be adapted to assess activity in neuronal processes. This has been greatly facilitated by the unusual topography of the GnRH neurons that results in the GnRH neuron projections becoming highly concentrated in the ventrolateral ARN before passing into the ME. The dendrons in this area display substantial rises in intracellular calcium in response to propagating action potentials as well as to the local application of kisspeptin (*Iremonger et al., 2017*; *Liu et al., 2021*). The locations of successful optic fibers immediately above and sometimes lateral to the ARN make it very unlikely that recordings were made of GnRH neuron activity at the level of their terminals within the ME.

The GnRH neuron dendrons exhibited clustered high-frequency (approximately 0.01 Hz) baseline activity that did not change in profile across 24 hr in males or throughout the estrous cycle. This would be predicted to give rise to fluctuating, low-level inter-pulse GnRH secretion perhaps compatible with observations made using high-frequency portal sampling in the ewe (*Evans et al., 1995*). High-frequency dopamine release also occurs within the ME between events determining prolactin secretion (*Romanò et al., 2017*). The physiological significance of inter-pulse GnRH release at the gonadotroph is poorly understood (*Le Tissier et al., 2017*). While the self-priming action of GnRH at the gonadotroph is well characterized for the surge, it remains unclear if this exists in relation to pulses (*Fink, 1995*). Indeed, optogenetic experiments driving GnRH neuron firing at high-frequency burst-like intervals in ovariectomized mice were not found to generate any appreciable change in LH secretion in vivo (*Campos and Herbison, 2014*).

The origin of fluctuating baseline activity in male and female GnRH neuron dendrons is unknown but it is tempting to speculate that it may arise from synchronised episodic activity of the GnRH neurons themselves. Brain slice electrophysiological studies have consistently documented repetitive burst firing in GnRH neuron cell bodies (*Herbison, 2015*; *Constantin et al., 2021a*), and similar patterns of activity were observed in GnRH neurons in anesthetized mice (*Constantin et al., 2013*). However, the clear clustered patterns of this activity recorded at the dendron would require at least some GnRH neurons to synchronize their activity. To date, there has been no evidence in support of this with dual or multiple recordings of GnRH neuron cell bodies in brain slices failing to show any coordinated activity (*Constantin et al., 2012*; *Chen and Moenter, 2023*). Hence, it is possible that the rapid high-frequency baseline activity is driven by inputs to GnRH neurons and, in this respect, it is interesting to note that the ARN kisspeptin neuron pulse generator also exhibits variable high-frequency activity between their synchronizations that drive GnRH pulses (*Han et al., 2023*).

We find abrupt increases in GnRH neuron activity preceding each LH pulse that rise to a peak in ~30 s and then dissipate over the following 8–10 min. This dynamic is very similar to the reported profile of GnRH secretion into the portal circulation of ovariectomised sheep (*Moenter et al., 1992b*). The abrupt nature of GnRH neuron activation is also paralleled by ARN kisspeptin neuron activity during each synchronization event that similarly rises to a peak within 25–50 s but then only lasts for a further ~1 min (*Han et al., 2019*; *McQuillan et al., 2019*). Kisspeptin released from terminals in the ventrolateral ARN operates through volume transmission to rapidly activate and synchronize GnRH neuron dendrons (*Iremonger et al., 2017*; *Liu et al., 2021*). Alongside evidence that pulsatile LH secretion is abolished in the absence of kisspeptin signaling (*Liu et al., 2021*), it is almost certain that the abrupt episodic GnRH neuron activation recorded here arises from up-stream synchronized firing of ARN kisspeptin neurons. What remains curious, however, is that the effects of exogenous kisspeptin on the dendron in vitro are typically prolonged lasting for over 30 minutes (*Iremonger et al., 2017*; *Liu et al., 2021*). This does not appear to be the case in vivo where GnRH neuron activity subsides within 10 min. It is possible that the termination of GnRH neuron dendron firing to provide a relatively constrained secretory signal involves other local mechanisms such as nitric oxide signaling (*Constantin et al., 2021a*).

Investigators using a variety of pituitary portal GnRH assays in rodent, sheep and primate models have suggested that GnRH neurons generate the LH surge through either a constant increase in activity or by increasing their normal pulsatile pattern of release (*Sarkar et al., 1976*; *Caraty et al., 1989*; *Moenter et al., 1992a*, *Xia et al., 1992*; *Clarke, 1993*; *Pau et al., 1993*; *Sisk et al., 2001*). We demonstrate here that, both predictions were correct with GnRH neuron activity changing in a multi-faceted manner comprised of a slowly oscillating baseline increase with superimposed episodic activity. Retrospectively, this is now readily discernable in many of those portal bleeding data (*Sarkar et al., 1976*; *Caraty et al., 1989*; *Moenter et al., 1992a*, *Xia et al., 1992*; *Clarke, 1993*; *Pau et al., 1993*; *Sisk et al., 2001*). The slow oscillating increase in baseline activity observed here is most probably the key event driving the LH surge. It initiates at the time the LH surge begins and has a duration of approximately 12 hr which is very similar to that of GnRH release in the portal system of rats, sheep, and monkeys (*Sarkar et al., 1976*; *Pau et al., 1993*; *Karsch et al., 1997*). The reason for the extremely prolonged period of GnRH neuron activity and secretion long outlasting the LH surge remains unknown but could be related to sexual behavior (*Pfaff, 1973*; *Skinner and Caraty, 2002*).

It has been surprising to find that the gradual surge increase in activity has a slow hourly oscillatory rhythm. While a circadian contribution to the onset of the surge is well established (*Tonsfeldt et al., 2022*), it had not been appreciated that an ultradian rhythm might also exist. It is very likely that the slow increase in GnRH activity recorded here is driven by the preoptic surge generator (*Wang et al., 2020*), and it will be interesting in future studies to establish whether ultradian rhythms exist within this circuitry and what function they may play in surge generation.

The 'two-compartment model' of GnRH neuron function involves independent pulse and surge generators that operate on different parts of the GnRH neuron to bring about pulsatile and surge patterns of gonadotropin secretion (*Herbison, 2020*). We have recently shown that selective suppression of preoptic area kisspeptin expression abolishes the surge generator but has no impact on pulsatile LH secretion (*Clarkson et al., 2023*). This model predicts that, on the afternoon of proestrus, the surge generator would operate coincidently with the pulse generator until such time as post-ovulatory progesterone secretion suppresses the pulse generator (*Herbison, 2020*). This is in perfect agreement with our present in vivo recordings where we find abrupt episodic pulse activity to continue on top of the rising phase of the baseline increase until the early hours of estrus when it stops. This highly stereotypical pattern of abrupt episodic activity that reduces in frequency across surge onset is clearly observable after deconvolution of the GCaMP signal (*Figure 6*) and is identical to that of the ARN kisspeptin pulse generator (*McQuillan et al., 2019*).

An important question is whether pulse generator activity at the time of the surge may contribute to ovulation. While pulse generator events are short in duration, they nevertheless increase the amplitude of total GnRH neuron activity during the rising phase of the surge and presumably also GnRH secretion within the portal system. Mice and sheep with a knockdown of ARN kisspeptin neurons have recently been reported to have reduced LH surge amplitude (*Aerts et al., 2024*; *Velasco et al., 2023*) consistent with the concept proposed here that the pulse generator contributes to the LH surge directly through activation of the GnRH neuron dendron. Intriguingly however, toxin-induced knockdown of ARN kisspeptin neurons in rats has been reported to have the opposite effect of increasing LH surge amplitude (*Helena et al., 2015*; *Mittelman-Smith et al., 2016*).

In summary, we provide here the first direct recordings of GnRH neuron activity in vivo. These observations demonstrate that alongside a baseline pattern of activity of unknown function, abrupt episodic activity underlies pulse generation whereas a slow and prolonged ultradian oscillation on proestrus is responsible for the preovulatory gonadotropin surge. The neurobiological mechanisms underlying this unexpected pattern of surge activity remain to be discovered.

## Materials and methods
### Animals
Male and female 129S6Sv/Ev C57BL/6 *Gnrh1*[Cre/+] mice (*Yoon et al., 2005*) crossed on to the Ai162 (TIT2L-GC6s-ICL-tTA2)-D Cre-dependent GCaMP6s line (JAX stock #031562) (*Daigle et al., 2018*) were group-housed in conventional cages with environmental enrichment under conditions of controlled temperature (22 ± 2°C) and lighting (12 hr light/12 hr dark cycle; lights on at 07:00) with ad libitum access to food (RM1-P, SDS, UK) and water. All animal experimental protocols were approved

by the University of Cambridge Animal Welfare and Ethics Review Body under the UK Home Office license, P174441DE.

## Stereotaxic implantation of optic fibers

Adult mice were anaesthetized with 2% isoflurane and placed in a stereotaxic frame with concurrent buprenorphine (0.05 mg/kg, s.c.) and meloxicam (5 mg/kg, s.c.) analgesia. Dexamethasone (10 mg/kg, s.c.) was used to prevent cranial swelling. A single 400 μm diameter optic fiber (0.48 NA, Doric lenses, QC, Canada) was implanted into the brain with the tip placed immediately above the dorso-medial part of ARN (A-P to bregma, –2.0 mm; D-V 5.9 mm, M-L ± 0.2 mm to sagittal sinus). Following 1 wk of surgery, all animals were handled daily and habituated to a photometry recording setup for at least 3 wk.

## GCaMP6 fiber photometry and blood sampling

Four to 12 wk following surgery, fiber photometry experiments were undertaken to record GCaMP fluorescence signal in freely behaving mice for 6 or 24 hr periods using a previously described methodology (*Han et al., 2019*; *Han et al., 2023*). This included a custom-built photometry system using Doric components (Doric Lenses, QC, Canada) and a National Instrument data acquisition board (TX, USA) based on a previous design (*Lerner et al., 2015*). Blue (465–490 nm) and violet (405 nm) LED lights were sinusoidally modulated at frequencies of 531 and 211 Hz, respectively and were focused onto a single fiber optic connected to the mouse. The light intensity at the tip of the fiber was 30–80 microwatts. Emitted fluorescence signal from the brain was collected via the same fiber, passed through a 500–550 nm emission filter, and focused onto a fluorescence detector (Doric, QC, Canada). The emissions were collected at 10 Hz and the two GCaMP6 emissions were recovered by demodulating the 465–490 nm signals (calcium-dependent) and 405 nm (calcium-independent) signals. Signals were either recorded in a continuous mode or a scheduled 5 s on/10 s off mode.

Analysis was performed in MATLAB with the subtraction of the 405 signal from the 465 -490 signal to extract the calcium-dependent signal followed by an exponential fit algorithm used to correct for baseline shift. The signal was converted to ΔF/F (%) values using the equation $\Delta F/F = ((F_{recorded} - F_{baseline})/F_{baseline}) \times 100$. The Findpeaks algorithm was used to detect dSEs, and the duration and the time of SE to the half-width full maximum were determined. For the deconvolution of signals from 24 hr proestrus recordings, the movmean algorithm was used to extract a 30 min rolling average.

With this approach, slow oscillatory shifts in baseline during the proestrous surge were detected. The onset and offset of the surge signals were determined by the points at which there was a>5% increment in ΔF relative to all previous timepoints, and when the calcium signal returned to 90% of the baseline value, respectively. The highest value observed during the slow oscillatory phase of the surge was determined to be the peak, and FWHM values were determined from the onset, peak, and offset values. The time taken between one trough to another in individual oscillations was calculated as the duration of each oscillation. The second phase of the deconvolution detected all dSE signals using the 'findpeaks' algorithm and signals between –60–360 s around each peak were separated to visualize the remaining low amplitude 'residual' signal. A threshold of 5% above baseline was used to extract the residual signal. Residual signals with peaks occurring >420 s from the preceding peak were considered to represent separate clusters.

To examine the relationship between calcium episodes with LH pulses, freely behaving mice were attached to the fiber photometry system, and 4 μL blood samples were obtained every 5–10 min from the tail tip over a period of 120–240 min. To assess the relationship between the long calcium increment during proestrus evening and the LH surge, female mice were attached to the fiber photometry system in the morning of proestrus, and blood samples (3 μL) were collected every 3 hr for 18 hr until the morning of estrus. Levels of LH were measured by in-house LH ELISA (*Steyn et al., 2013*) with an assay sensitivity of 0.04 ng/mL and intra-assay coefficient of variation of 8.2%.

## Immunohistochemistry

Adult GnRH-Cre, GCaMP6s mice were given a lethal overdose of pentobarbital (3 mg/100 μL, i.p.) and perfused transcardially with 4% paraformaldehyde. Brains were processed for dual GFP and GnRH immunofluorescence. For GFP immunostaining, anti-chicken GFP (1:5000, Aves Lab) was used followed by AlexaFluor 488-conjugated goat-anti-chicken (1:1000). For GnRH cell body immunostaining, GA2

guinea pig anti-GnRH antisera (1:3000, gift from G.Anderson, New Zealand) was used in combination with AlexaFluor 647-conjugated goat anti-guinea pig immunoglobulin (1:500, Thermo Fisher Scientific, USA). For GnRH dendron immunostaining, rabbit anti-GnRH (1:20,000, LR1, gift of R.Benoit, Montreal) antisera were used followed by biotinylated goat anti-rabbit immunoglobulin (1:1000, Jackson Immunoresearch) and AlexaFluor 568-conjugated Streptavidin (1:400, Thermo Fisher Scientific, USA). Imaging was performed using a Leica SP8 Laser Scanning Confocal Microscope (Leica Microsystems) at the Cambridge Advanced Imaging Center and analyzed using ImageJ.

## Statistical analysis

All statistical analyses were performed in Prism 10 (GraphPad Software Inc). All values given in this study are mean ± SEM, and significance is defined as $p < 0.05*$ or $p < 0.01**$. For inter-peak interval analysis in females across the estrous cycle and the residual cluster activity analysis, Kruskal Wallis ANOVA followed by Dunn's post-hoc tests was used.

## Acknowledgements

This work was supported by the Wellcome Trust (212242/Z/18/Z). JCK received a Doc. Mobility Fellowship (P1ZHP1_184166) from the Swiss National Science Foundation. We thank Ms. Maria Pardo-Navarro for technical assistance. For the purpose of open access, the author has applied a Creative Commons Attribution (CC BY) license to any Author Accepted Manuscript version arising from this submission.

## Additional information

### Funding

| Funder | Grant reference number | Author |
| --- | --- | --- |
| Wellcome Trust | 10.35802/212242 | Allan E Herbison |
| National Science Foundation | P1ZHP1_184166 | Jae-Chang Kim |

The funders had no role in study design, data collection and interpretation, or the decision to submit the work for publication. For the purpose of Open Access, the authors have applied a CC BY public copyright license to any Author Accepted Manuscript version arising from this submission.

### Author contributions

Su Young Han, Data curation, Formal analysis, Investigation, Methodology, Writing – review and editing; Shel-Hwa Yeo, Formal analysis, Investigation; Jae-Chang Kim, Software; Ziyue Zhou, Investigation; Allan E Herbison, Conceptualization, Resources, Supervision, Funding acquisition, Methodology, Writing – original draft, Project administration, Writing – review and editing

### Author ORCIDs

Jae-Chang Kim ⓘ https://orcid.org/0000-0001-5224-817X
Ziyue Zhou ⓘ https://orcid.org/0000-0003-4725-7543
Allan E Herbison ⓘ https://orcid.org/0000-0002-9615-3022

### Ethics

All animal handling and experimental protocols were undertaken as approved by the Animal Welfare and Ethical Review Body of the University of Cambridge under UK Home Office project license P174441DE. All surgery was performed under isofluorane anesthesia, and every effort was made to minimize suffering.

Reviewer #1 (Public review): https://doi.org/10.7554/eLife.100856.3.sa1
Reviewer #2 (Public review): https://doi.org/10.7554/eLife.100856.3.sa2
Author response https://doi.org/10.7554/eLife.100856.3.sa3

## Data availability

All data analysed during this study are included in the manuscript and source data files have been uploaded to the Dryad Digital Repository.

The following dataset was generated:

| Author(s) | Year | Dataset title | Dataset URL | Database and Identifier |
| --- | --- | --- | --- | --- |
| Han S, Yeo S, Kim J, Herbison AE | 2024 | Data from: Multi-dimensional oscillatory activity of mouse GnRH neurons in vivo | https://doi.org/10.5061/dryad.pnvx0k6zr | Dryad Digital Repository, 10.5061/dryad.pnvx0k6zr |

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
