## [Editor Report · eLife Assessment]

This **valuable** study investigates the oscillatory activity of gonadotropin-releasing hormone (GnRH) neurones in mice using GCaMP fiber photometry. It demonstrates three distinct patterns of oscillatory activity that occur in GnRH neurons comprising low-level rapid baseline activity, abrupt short-duration oscillations that drive pulsatile gonadotropin secretion, and, in females, a gradual and prolonged oscillating increase in activity responsible for the relatively short-lived preovulatory LH surge. The evidence presented in the study is **solid**, offering theoretical implications for understanding the behaviour of GnRH neurones in the context of reproductive physiology, and will be of interest to researchers in neuroendocrinology and reproductive biology.

---

## [Referee Report · Reviewer #1 (Public review)]

Summary:

The authors aimed to investigate the oscillatory activity of GnRH neurones in freely behaving mice. By utilising GCaMP fiber photometry, they sought to record real-time neuronal activity to understand the patterns and dynamics of GnRH neuron firing and their implications for reproductive physiology.

Strengths:

- The use of GCaMP fiber photometry allows for high temporal resolution recordings of neuronal activity, providing real-time data on the dynamics of GnRH neurones.

- Recording in freely behaving animals ensures that the findings are physiologically relevant and not artifacts of a controlled laboratory environment.

- The authors used statistical methods to characterise the oscillatory patterns, ensuring the reliability of their findings.

Weaknesses:

- While the study identifies distinct oscillatory patterns in GnRH neurones' calcium dynamics, it falls short in exploring the functional implications of these patterns for GnRH pulsatility and overall reproductive physiology.

- The study lacks broader discussion to include comparisons with existing studies on GnRH neurone activity and pulsatility and highlight how the findings of this study align with or differ from previous research and what novel contributions are made.

- The authors aimed to characterise the oscillatory activity of GnRH neurons and successfully identified distinct oscillatory patterns. The results support the conclusion that GnRH neurons exhibit complex oscillatory behaviours, which are critical for understanding their role in reproductive physiology. However, it has not been made clear what exactly do the authors mean by "multi-dimensional oscillatory patterns" and how has this been shown.

---

## [Referee Report · Reviewer #2 (Public review)]

Summary:

In this manuscript, the authors report GCaMP fiber-photometry recordings from the GnRH neuron distal projections in the ventral arcuate nucleus. The recording are taken from intact, male and female, freely behaving mice. The report three patterns of neuronal activity:

1. abrupt increases in the Ca2+ signals that are perfectly correlated with LH pulses.

2. a gradual, yet fluctuating (with a slow ultradian frequency), increase in activity, which is associated with the onset of the LH surge in female animals.

3. clustered (high frequency) baseline activity in both female and male animals.

Strengths:

The GCaMP fiber-photometry recordings reported here are the first direct recordings from GnRH neurones in free behaving mice. These recordings suggest a rich repertoire of activity, including the integration of distinct "surge" and "pulse" generation signals, and an ultradian rhythm during the onset of the surge.

Weaknesses:

The data analysis methods used for the characterisation of the oscillatory behaviour could be complemented with more advanced wavelet methods to quantify and analyse how the frequency content of the observed Ca2+ signal changes over the cycle.

---

## [Author Response]

The following is the authors’ response to the original reviews.

**Public Reviews:**

**Reviewer #1 (Public Review):**
Summary:The authors aimed to investigate the oscillatory activity of GnRH neurones in freely behaving mice. By utilising GCaMP fiber photometry, they sought to record real-time neuronal activity to understand the patterns and dynamics of GnRH neuron firing and their implications for reproductive physiology.Strengths:(1) The use of GCaMP fiber photometry allows for high temporal resolution recordings of neuronal activity, providing real-time data on the dynamics of GnRH neurones.(2) Recording in freely behaving animals ensures that the findings are physiologically relevant and not artifacts of a controlled laboratory environment.(3) The authors used statistical methods to characterise the oscillatory patterns, ensuring the reliability of their findings.Weaknesses:(1) While the study identifies distinct oscillatory patterns in GnRH neurones' calcium dynamics, it falls short in exploring the functional implications of these patterns for GnRH pulsatility and overall reproductive physiology.

The functional roles of pulsatile and surge patterns of GnRH release are extremely well established. We have found perfect correlations between GnRH neuron dendron GCaMP activity and LH pulses as well as the LH surge clearly indicating the function of these activity patterns. We do not know the functional role of the clustered high-frequency basal activity that we have discovered and, as noted in the Discussion, are unsure of its physiological importance. Although it may be minor, it will require future investigation.

(2) The study lacks a broader discussion to include comparisons with existing studies on GnRH neurone activity and pulsatility and highlight how the findings of this study align with or differ from previous research and what novel contributions are made.

The Reviewer fails to recognise that these are first recordings of GnRH neurons in vivo. There are no prior studies for comparison. We have noted the only other in vivo study (undertaken by ourselves) many years ago in anaesthetized mice. It was never expected that electrophysiological recordings of GnRH neurons in acute brain slices (by ourselves and others) would reflect their activity in vivo. Now that we know this to be the case, it would be churlish to point this out explicitly. We have made some modifications to the Discussion by comparing the present data more thoroughly with other in vivo GnRH secretion and kisspeptin neuron activity studies.

(3) The authors aimed to characterise the oscillatory activity of GnRH neurons and successfully identified distinct oscillatory patterns. The results support the conclusion that GnRH neurons exhibit complex oscillatory behaviours, which are critical for understanding their role in reproductive physiology. However, it has not been made clear what exactly the authors mean by "multi-dimensional oscillatory patterns" and how has this been shown.

The study shows three types of GnRH neuron activity; two of which would be classified as oscillatory in nature and these show different temporal dimensions.

**Reviewer #2 (Public Review):**
Summary:In this manuscript, the authors report GCaMP fiber-photometry recordings from the GnRH neuron distal projections in the ventral arcuate nucleus. The recordings are taken from intact, male and female, freely behaving mice. The report three patterns of neuronal activity:(1) Abrupt increases in the Ca2+ signals that are perfectly correlated with LH pulses.(2) A gradual, yet fluctuating (with a slow ultradian frequency), increase in activity, which is associated with the onset of the LH surge in female animals.(3) Clustered (high frequency) baseline activity in both female and male animals.Strengths:The GCaMP fiber-photometry recordings reported here are the first direct recordings from GnRH neurones in vivo. These recordings have uncovered a rich repertoire of activity suggesting the integration of distinct "surge" and "pulse" generation signals, and an ultradian rhythm during the onset of the surge.Weaknesses:The data analysis method used for the characterisation of the ultradian rhythm observed during the onset of the surge is not detailed enough. Hence, I'm left wondering whether this rhythm is in any way correlated with the clusters of activity observed during the rest of the cycle and which have similar duration.

We have provided further information on the characterisation of the ultradian rhythm observed at the time of the surge. Whether this is related to the clustered basal activity is an interesting point but very difficult to resolve. We note that the “basal” and “surge” ultradian oscillations have very different durations of ~30 and ~80 min suggesting that they may be independent phenomenon. However, the only way to really exclude a similar genesis will be to establish the origin of each type of oscillatory activity. Preliminary data in the lab show that the RP3V kisspeptin neurons exhibit an identical pattern of ultradian oscillation at the time of the surge leading us to suspect that the surge oscillation is driven by this input. As noted in the Discussion it is presently difficult to determine where the high basal activity originates.

**Recommendations for the authors:**

**Reviewer #1 (Recommendations For The Authors):**
(1) Evidence of Multi-Dimensional Oscillatory Patterns: The manuscript presents data showing the oscillatory activity of GnRH neurones with distinct frequency and amplitude characteristics. The analysis includes statistical tests that illustrate the variability in neuronal firing patterns. However, the multi-dimensional nature of this activity has not been demonstrated. It is not clear what is meant by "dimension" with regard to the calcium recordings (oscillatory activity). If the authors refer to the frequency content of the calcium signal then a proper Fourier or Wavelet analysis should be carried out to characterise the multiple frequencies present in the calcium dynamics in male mice and during various stages of the cycle in female mice

The study shows three types of GnRH neuron activity; two of which would be classified as oscillatory in nature. One occurs for ~10 min every hour or so and the other occurs for ~ 12 hours once every 4-5 days. This does not require any analysis to distinguish between the two or claim that they are different i.e. multidimensional.

(2) Data Interpretation: Expand the discussion on the physiological relevance of the identified oscillatory patterns. Specifically, explore how these patterns might influence GnRH pulsatility, hormone secretion dynamics, and reproductive cycles.

The functional roles of pulsatile and surge patterns of GnRH release are extremely well established. We have found perfect correlations between GnRH neuron dendron GCaMP activity and LH pulses as well as the LH surge clearly indicating the function of these activity patterns. We do not know the functional role of the clustered high-frequency basal activity that we have discovered and, as noted in the Discussion, are unsure of its physiological importance. Although it may be minor, it will require future investigation.

(3) Literature Contextualisation: Broaden the discussion to include comparisons with existing studies on GnRH neuron activity and pulsatility. Highlight how the findings of this study align with or differ from previous research and what novel contributions are made.

The Reviewer fails to recognise that these are first recordings of GnRH neurons in vivo. There are no prior studies for comparison. We have noted the only other in vivo study (undertaken by ourselves) many years ago in anaesthetized mice. It would be naive to expect that electrophysiological recordings of GnRH neurons in acute brain slices (by ourselves and others) would reflect their activity in vivo. Now that we know this to be the case, it would be churlish to point this out explicitly. We have made some modifications to the Discussion by comparing the present data more thoroughly with other in vivo GnRH secretion and kisspeptin neuron activity studies.

(4) Future Directions: Suggest potential follow-up experiments to explore the regulatory mechanisms underlying the observed oscillatory patterns. This could include investigating the role of neurotransmitters, hormonal feedback mechanisms, and other factors that might influence GnRH neuron activity.By addressing these recommendations, the authors can further strengthen their manuscript and enhance its impact on the field.
**Reviewer #2 (Recommendations For The Authors):**
Suggestions:(1) The authors might want to analyse their inter-peak interval data by fitting them to a simple parametric statistical model (the gamma distribution would be a good choice to capture the skewness of these data). This way they would be able to describe the observed variability, and if the fits are not good back up to their claims "The dSEs occurred on average ... and showed no clear modal distribution pattern (Fig. 2D)".

Thank you for the suggestion. We have carried out Shapiro-Wilk tests for male inter-peak interval distribution and found a W value of 0.87 and P value <0.0001****, providing strong evidence that the data is not normally distributed. Skewness and Kurtosis values are 1.39 and 1.81 respectively, indicating that the distribution is right-skewed with a platykurtic distribution, indicating that the data is less peaked and more spread out than the normal distribution (with a kurtosis of 3). This has now been added to the manuscript.

(2) If I understand correctly, in Figure 3D, inter-peak intervals from all 4 stages of the estrus cycle are pooled together. It would also be interesting if the authors gave the interval histograms for the different stages of the cycle separately.

We have now plotted the inter-peak interval distribution histograms for each individual cycle next to the example traces in Figure 3. The descriptions of the distribution pattern are also updated in the figure legends.

(3) In Figure 3C, one can see the mean interval for different animals (as open circles), is that right? Is the statistical test run on these animals mean, or is the entire dSEs dataset used? In any case, it's not clear to the reader how variable intervals are in individual recordings from each animal. Could the authors add this information (could be easily added in the figure caption)?

The reviewer is correct, that each open circle is the mean interval for each animal. The statistical test was run on the animals mean. Now this information is added to the figure legend.

(4) The authors should explain how they identify the regions (clusters) of high-frequency baseline activity, which they present in Figure 4.

The relevant information is now added to the methods section under the heading ‘GCaMP6 fiber photometry and blood sampling’.

(5) The authors should detail how to identify and characterise the ultradian rhythm they observe at the onset of the surge.

The relevant information is now added to the methods section under the heading ‘GCaMP6 fiber photometry and blood sampling’.

(6) The author could perform some kind of wavelet-type analysis to quantify and analyse how the frequency content of the observed Ca2+ signal changes over the cycle. From their current analysis, I am not sure whether the ultradian oscillations they observe during the surge are related to the low-activity cluster events they observe during the other stages of the cycle.

This is an interesting point but very difficult to resolve. We note that the “basal” and “surge” ultradian oscillations have very different durations of ~30 and ~80 min suggesting that they may be independent phenomenon. However, the only way to really exclude a similar genesis will be to establish the origin of each type of oscillatory activity. Preliminary data in the lab show that the RP3V kisspeptin neurons exhibit an identical pattern of ultradian oscillation at the time of the surge leading us to suspect that the surge oscillation is driven by this input. As noted in the Discussion it is presently difficult to determine where the high basal activity originates.